# Individual and Community-Level Risk Factors for Giardiasis in Children under Five Years of Age in Pakistan: A Prospective Multi-Regional Study

**DOI:** 10.3390/children10061087

**Published:** 2023-06-20

**Authors:** Irfan Khattak, Wen-Lan Yen, Tahir Usman, Nasreen Nasreen, Adil Khan, Saghir Ahmad, Gauhar Rehman, Khurshaid Khan, Mourad Ben Said, Chien-Chin Chen

**Affiliations:** 1College of Veterinary and Animal Husbandry, Abdul Wali Khan University, Mardan 23200, Pakistan; irfankhattak@awkum.edu.pk (I.K.); tahircau@gmail.com (T.U.); 2Department of Pediatrics, Ditmanson Medical Foundation Chia-Yi Christian Hospital, Chiayi 60002, Taiwan; 07066@cych.org.tw; 3Department of Zoology, Abdul Wali Khan University, Mardan 23200, Pakistan; nasreen@awkum.edu.pk (N.N.); gauhar@awkum.edu.pk (G.R.); khurshaidkhan@awkum.edu.pk (K.K.); 4Department of Zoology, Bacha Khan University, Charsadda 24540, Pakistan; 5Department of Biology, Mount Allison University, Sackville, NB E4L1E4, Canada; 6Department of Microbiology, Abdul Wali Khan University, Mardan 23200, Pakistan; saghir@awkum.edu.pk; 7Laboratory of Microbiology, National School of Veterinary Medicine of Sidi Thabet, University of Manouba, Manouba 2010, Tunisia; bensaidmourad83@yahoo.fr; 8Department of Basic Sciences, Higher Institute of Biotechnology of Sidi Thabet, University of Manouba, Manouba 2010, Tunisia; 9Department of Pathology, Ditmanson Medical Foundation Chia-Yi Christian Hospital, Chiayi 60002, Taiwan; 10Department of Cosmetic Science, Chia Nan University of Pharmacy and Science, Tainan 71710, Taiwan; 11Ph.D. Program in Translational Medicine, Rong Hsing Research Center for Translational Medicine, National Chung Hsing University, Taichung 40227, Taiwan; 12Department of Biotechnology and Bioindustry Sciences, College of Bioscience and Biotechnology, National Cheng Kung University, Tainan 70101, Taiwan

**Keywords:** children, giardiasis, *Giardia lamblia*, immunology, infection, pediatric, public health, risk factors, seroprevalence

## Abstract

Objectives: This study aimed to estimate the prevalence of *Giardia lamblia* infection and identify associated risk factors at both individual and community levels in a pediatric population in different agroecological zones of Khyber Pakhtunkhwa, Pakistan. Methods: A community-based cross-sectional study was conducted from March to December 2022. Using stratified sampling, 1026 households were recruited from nine agroecological zones. Stool samples were collected from 1026 children up to the age of five years and processed for detection of Giardia using a commercial ELISA kit. Data on potential risk factors were collected using a pre-structured questionnaire. A multivariable logistic regression model was used to identify risk factors associated with giardiasis. Results: The study found that the prevalence of giardiasis in the study area was 3.31%. Children aged 13–24 months were found to be at higher risk for giardiasis. Illiterate mothers and fathers attending daycare institutions/kindergartens, mothers not practicing hand washing during critical times, households with companion animals, and homes where stray dogs/cats enter were identified as predictors of giardiasis at the individual level. Children living in sub-mountain valleys use un-piped water, inadequate domestic water storage vessels, drink un-boiled or unfiltered water, live near rubbish heaps or un-paved streets/pathways, and have unimproved latrine facilities were identified as risk factors of giardiasis at the community level. Conclusions: The study highlights the need for integrated intervention approaches at both individual and community levels to reduce the incidence of giardiasis in Khyber Pakhtunkhwa, Pakistan. Interventions aimed at promoting behavioral change and providing safe and adequate water sources, combined with individual-level interventions such as hand washing and awareness of giardiasis prevention methods, would be critical to addressing this health concern. Inter-sectoral collaboration between the health sector and other sectors would also be necessary to achieve meaningful progress in reducing the incidence of giardiasis in resource-limited areas.

## 1. Introduction

*Giardia lamblia* (*G. lamblia*) is the most common protozoan to infect the small intestine in humans, especially children [1]. Giardiasis is a major cause of enteric infections globally, with a prevalence of 10% to 50% of the population in developing countries [2,3]. It is estimated that approximately 280 million people worldwide have symptomatic giardiasis and around 500,000 new cases are reported annually [4]. *G. lamblia* is the only species that infects humans and other mammals [5], and it is classified into seven different genotypes/assemblages (A to G), each specific to a particular host. Assemblages A and B have been identified in humans, cattle, and various other mammals [6].

*Giardia lamblia* has two life cycle stages: trophozoite and cyst (Figure 1). Once ingested, the cysts undergo excystation in the proximal small intestine, which releases trophozoites. These trophozoites then attach to the walls of the small intestine and rapidly multiply, leading to clinical illness in humans [7]. Giardia cysts are the parasite’s environmentally stable stage and can survive for up to two months without being inactivated by common water disinfectants [6]. Contaminated food and water are the primary means of Giardia transmission. Other contributing factors include poor living conditions, overcrowded housing, inadequate environmental sanitation, and unclean personal hygiene practices.

*G. lamblia* can cause a range of clinical symptoms, from asymptomatic to acute or chronic diarrhea with malabsorption syndrome and weight loss [8]. Giardiasis often presents with diarrhea, lethargy, greasy feces, gas, abdominal cramping, bloating, and weight loss [9]. *G. lamblia* also significantly contributes to recurrent abdominal pain in children [10]. In addition, the prevalence of intestinal parasites, including *Giardia*, is influenced by various socioeconomic factors such as poverty, inadequate sanitation and water treatment systems, illiteracy, poor hygienic practices, and proximity to domestic and wild animals. These factors increase the risk of *Giardia* infection in different communities [11,12].

Few studies have investigated the prevalence and risk factors associated with *Giardia* infection in Pakistan in the past 20 years, with most studies focusing solely on children [7,13]. Reports indicate that between 9% and 10% of Pakistan’s population has *G. lamblia*, with malnourished children under 10 years of age particularly affected [14].

Although numerous studies have been conducted on the prevalence and distribution of giardiasis in Pakistan, there are still regions where epidemiological data are lacking, including Khyber Pakhtunkhwa. This study represents the first attempt to estimate the frequency of *G. lamblia* infection and identify contributing factors in this region. The study aims to determine the prevalence of *G. lamblia* infection in a healthy pediatric population across several Agro-ecological Zones (AEZs) of Khyber Pakhtunkhwa, Pakistan, and to identify any potential individual and community-level risk factors.

## 2. Materials and Methods

### 2.1. Sampling Technique and Sample Size Calculation 

From March to December 2022, a cross-sectional study was conducted in the Khyber Pakhtunkhwa Province of Pakistan to determine the risk factors associated with giardiasis in children under five at both the individual and community levels. The province encompasses nine AEZs, including High Dry Mountains, Sub-Mountain Valleys, Sub-Humid Mountains, Wet Mountains, Valley Plains, Piedmont Plains, Semi-Arid Piedmont, Western Mountains, and Desert Plains. Each zone was treated as a stratum, and the stratified sampling technique was employed to select the study participants to collect an equal number of samples from each AEZ and ensure they were representative of their respective AEZs. Based on the previously reported prevalence of 8.0% [15], the sample size for each AEZ (stratum) was calculated to be 114, resulting in a total sample size of 1026 for all nine AEZs.

### 2.2. Stool Sample Collection and Giardia Detection

Random households were selected for both stool and data collection. If a household had more than one eligible child (under the age of five), one child was chosen using a lottery method. A total of 1026 stool samples were collected from the selected children and transported in an icebox to the Preventive Medicine and Public Health Laboratory at Abdul Wali Khan University Mardan for further processing. *Giardia* was detected in the stool samples using a commercial ELISA kit (IBL International GmbH, Hamburg, Germany) following the manufacturer’s instructions. Negative and positive controls were added to the first and second wells, respectively, while 50 μL of each sample was added to the remaining wells. Dilution buffer was then added to each sample well, and the mixture was incubated for 60 min at a temperature of 15–25 °C. After incubation, the wells were washed seven times with wash buffer, and enzyme conjugate was added to each well (including control wells) and incubated for 30 min at 15–25 °C. The wells were then washed seven times with wash buffer again. Chromogen was added to each well and incubated for 10 min at 15–25 °C, followed by the addition of a stop solution. Finally, an ELISA reader was used to take the reading. 

### 2.3. Questionnaire and Observation Checklist for Assessing Giardiasis Risk Factors 

In this study, we employed a comprehensive questionnaire and observation checklist to assess the risk factors associated with giardiasis at both the individual and community levels [Appendix A]. The observation checklist was designed to gather data on various environmental factors, including domestic water storage vessels, the presence of rubbish heaps near homes, the condition of streets or pathways, latrine facilities, and the presence of companion animals. Simultaneously, the questionnaire focused on evaluating individual-level risk factors. It encompassed a range of factors, such as child gender, age, family size, mother’s and father’s educational level, attendance at daycare or kindergarten, handwashing behavior, and the potential entry of stray dogs and cats into the home. To ensure accuracy and reliability, face-to-face interviews were conducted with guardians or primary caregivers to gather data on these specific risk factors. Furthermore, community-level risk factors were also assessed as part of our investigation. These included the AEZs in the study area, drinking water sources, the usage of boiled or filtered drinking water, domestic water storage vessels, the presence of rubbish heaps near homes, the condition of streets or pathways, and the availability and quality of latrine facilities. By using this comprehensive questionnaire and observation checklist, we aimed to gather a detailed understanding of the risk factors associated with giardiasis at both the individual and community levels. This approach allowed us to capture a broad range of variables and factors that could potentially contribute to the prevalence of giardiasis in the studied population.

### 2.4. Data Processing and Statistical Analysis

The statistical software STATA version 16 was utilized to analyze the data with a significance level of 5% and a 95% confidence interval. Logistic regression models, including classical and multilevel methods, were employed to identify risk factors for giardiasis at the individual and community levels. The impact of potential risk factors on giardiasis occurrence was measured using multivariable logistic regression that considered individual and community-level factors. Four models were generated in this study to identify giardiasis risk factors in the study area. An initial model without any explanatory variables was employed to assess the variance of the outcome variable and served as a baseline for subsequent models. The model I and II were adjusted to include individual and community-level factors, respectively. The final full model included both individual and community-level factors. The relationship between the outcome variable (giardiasis) and explanatory variables (individual and community-level risk factors) was expressed using the adjusted odds ratio (AOR), and the significance level was set at *p* < 0.05 (two-tailed).

## 3. Results

### 3.1. Giardiasis Prevalences According to Individual-Level Factors in Children under Five

The research was conducted on 1026 stool samples collected from children under five to determine Giardia’s presence. Results showed that out of the samples tested, 34 tested positive, indicating a prevalence rate of 3.31%. The study also revealed that diarrhea was higher in male children (3.7%) than in female children. Moreover, the prevalence rate was 8% among children aged between 13 to 24 months and 3.4% among children from families with more than six members. The study further showed that the prevalence rate of giardiasis was 4.4% among children of illiterate mothers and 8.1% among children of illiterate fathers. Additionally, children who attended daycare institutions or kindergartens had an 8.8% prevalence rate of the disease. The study also found that 6.88% of giardiasis cases were reported among mothers who did not wash their hands during critical times, while 4.83% were observed among children from families with companion animals (Table 1). 

### 3.2. Giardiasis Prevalences According to Community-Level Factors in Children under Five 

The prevalence of giardiasis varies among regions, with the highest rates found in sub-mountain valleys (6.14%), high dry mountains (4.39%), and valley plains (4.39%). Lower rates were observed in sub-humid mountains (3.51%), wet mountains (3.51%), semi-arid piedmont (2.63%), western mountains (2.63%), piedmont plains (1.75%), and desert plains (0.88%). Among children who use un-piped water, 7.22% tested positive for giardiasis, while 3.47% of those who used un-boiled or unfiltered water had the infection. Children with inadequate domestic water storage vessels had a 9.27% prevalence of giardiasis, and those with a rubbish heap near their house had a prevalence estimated at 8.94%. Children with unpaved streets/pathways to their houses and those whose mothers used unimproved latrine facilities had a higher prevalence of giardiasis, at 4.08% and 8.41%, respectively (Table 2).

### 3.3. Significant Factors Associated with Giardiasis in Children under Five 

The results of the multivariable analyses indicated that children aged 13–24 months had a 3.92 times higher risk of giardiasis (AOR 3.92; 95% CI 1.82–7.71) compared to those aged 49–60 months. Moreover, children of illiterate mothers had a 2.22 times higher risk of giardiasis (AOR 2.22; 95% CI 1.11–4.52) than those whose mothers had higher education. Similarly, children of illiterate fathers had a 4.82 times higher risk (AOR 4.82; 95% CI 2.33–9.61) than those whose fathers had higher education. Children who attended daycare institutions/kindergartens had a 3.12 times higher risk of giardiasis (AOR 3.12; 95% CI 1.61–6.42) than those who did not. Moreover, children whose mothers did not practice hand washing during critical times had a 3.61 times higher risk of giardiasis (AOR 3.61; 95% CI 1.91–6.94) compared to those whose mothers did practice hand washing. The risk of giardiasis was 5.72 times higher (AOR 5.72; 95% CI 2.81–11.78) among children with companion animals compared to those without. However, the risk was 4.62 times higher (AOR 4.62; 95% CI 2.23–8.92) among children in homes where stray dogs/cats entered compared to those where they did not. Lastly, residents of Sub-mountain valleys had a 7.23 times higher risk of giardiasis (AOR 7.23, 95% CI 3.51–13.74) than those in the Desert plains. According to a study, children from families that use un-piped water are 3.30 times more likely to contract giardiasis than those who use piped water. Furthermore, the likelihood of children getting giardiasis increases by 6.5 times when their families have inadequate domestic water storage vessels. Children who consume un-boiled or unfiltered water are 2.88 times more likely to contract giardiasis than those who drink boiled or filtered water. Living near rubbish heaps increases the risk of getting giardiasis by 5.2 times, while living near unpaved streets or pathways increases the risk by 2.19 times. Finally, families with unimproved Latrine facilities are 4.53 times more likely to have children with giardiasis than those with improved Latrine facilities (Table 3 and Table 4).

## 4. Discussion

*G. lamblia* is a common pathogenic protozoan that infects the small intestine and is a leading cause of death among children in developing countries [16]. The severity of *G. lamblia* infections is more prevalent in children than adults due to malnutrition, growth retardation, and inadequate care [17]. Poor personal hygiene, unhygienic toilet practices, consumption of unwashed fruits and vegetables, and the consumption of contaminated water and food contribute to the high morbidity rates among school children [18]. The study aims to determine the frequency of *G. lamblia* infection in children in various regions of Khyber Pakhtunkhwa, Pakistan. The research also explores potential individual and community-level risk factors associated with Giardia infection. The study employed direct microscopic examination or an immunochromatographic test (ICT) for epidemiological investigation, with ICT being more cost-effective. However, the ICT test’s low sensitivity could produce misleading information about the disease’s prevalence.

The results of our study indicate that the prevalence of *G. lamblia* infection among students in selected elementary schools is 3.31%. Other studies conducted in Pakistan have reported higher rates of *G. lamblia* infection, including 12.5% in the Khanewal district [13] and 6.5% in Faisalabad [13]. In Peshawar, the prevalence rate was even higher at 30.5% [19]. In comparison, Afghan refugees had a significantly higher prevalence rate of 37.7% [20], while Guatemala had the highest prevalence rate of 43.8% [21]. These variations in the prevalence rates of *G. lamblia* infection can be attributed to the socioeconomic differences between countries. For example, industrialized countries typically report lower prevalence rates of 2–7%, while developing countries have rates that can reach up to 40% [4].

The study found several risk factors associated with a higher risk of giardiasis infection at the individual level. The results indicated that children aged between 13–24 months were at a higher risk of contracting giardiasis (95% CI for AOR 3.92; 1.82–7.71). This finding is consistent with earlier studies [13,22,23]. The increased risk for young children could be due to their lack of acquired immunity and their higher exposure to the source of infection, which is often linked to inadequate personal hygiene and crowded living conditions. Furthermore, the study revealed that children attending daycare institutions or kindergartens (95% CI for AOR 3.12; 1.61–6.42) were also at a higher risk of giardiasis infection due to their poor hygiene habits and exposure to contaminated conditions. Daycare centers are places where children are highly vulnerable to parasites through interpersonal contact and unsanitary conditions. This finding is in line with earlier studies [13,24,25,26]. Overall, these findings suggest that young children and those attending daycare institutions/kindergartens are more susceptible to giardiasis infections due to their inadequate personal hygiene practices and exposure to contaminated environments.

The current study found no significant difference between males and females in terms of infection rate. Both genders had similar infection rates, with males at 3.7% and females at 3.1%. This finding is consistent with previous research [23,27,28]. The prevalence of intestinal parasites in children can be attributed to several factors, including food quality, water supply, personal and community hygiene, climate, sanitation conditions, proximity to domestic and wild animals, and socioeconomic status [29].

Our study found that parents’ educational level is a risk factor for giardiasis. Specifically, our results showed that children with illiterate mothers (95% CI for AOR 2.22; 1.11–4.52) and illiterate fathers (95% CI for AOR 4.82; 2.33–9.61) were more likely to be infected with *G. lamblia*. This finding is consistent with previous studies conducted in different regions of Pakistan [13,30], Malaysia [31], Tehran [32], and Mexico [24]. Moreover, other research has shown that a father’s educational level is also linked to a higher risk of *G. lamblia* infection, possibly due to its association with lower socioeconomic status and inadequate hygiene and sanitation practices [33]. Therefore, it is important to consider the influence of parents’ educational level and socioeconomic status when implementing prevention and control strategies for giardiasis.

Effective personal hygiene practices, such as handwashing, are crucial in preventing fecal-oral infections, including those caused by protozoa. Giardia cysts, for instance, can survive in the environment for an extended period, making handwashing a critical intervention to limit transmission via caregivers’ hands and contaminated food-common routes of infection [34]. The current study’s results support handwashing’s effectiveness in reducing the risk of giardiasis. Specifically, the study found that mothers who did not wash their hands during critical times had a higher risk of giardiasis (95% CI for AOR 3.61; 1.91–6.94). These findings are consistent with the study by [35], which reported a significant reduction in Giardia prevalence and infection intensity through handwashing promotion.

Our study revealed that owning companion animals (adjusted odds ratio [AOR] 5.72; 95% confidence interval [CI] 2.81–11.78) and living in homes with stray dogs/cats (AOR 4.62; 95% CI 2.23–8.92) were associated with a high incidence of giardiasis and were identified as risk factors for *G. lamblia* infection. It is widely recognized that *G. lamblia* is an intestinal parasite that primarily affects domestic animals, such as livestock, dogs, and cats, and has been isolated in varying prevalence from livestock and companion animals worldwide. For instance, dairy calves have been found to have the highest frequency of *G. lamblia* infection [36,37,38,39,40], while pigs [41], sheep [42], goats [43], elks and deer [44], dogs [45], and other ruminants [46] have also been found to carry the parasite.

The current study identified several risk factors at the community level. Our findings indicate that the prevalence of *G. lamblia* is significantly higher among children who consume un-piped water (95% CI for AOR 3.30; 1.5–6.8), use inadequate domestic water storage vessels (95% CI for AOR 6.5; 3.06–12.88), or drink un-boiled or unfiltered water (95% CI for AOR 2.88; 1.46–6.12). Contaminated drinking water is a well-known cause of many bacterial and parasitic diseases. Giardia and Cryptosporidium are recognized as the most common causes of waterborne outbreaks globally [47]. Our results are consistent with those of previous studies [13,48,49]. However, according to other studies [33,50], the risk of infection increases four-fold if untreated water is used.

Our research revealed that individuals living in houses without functioning toilets had higher chances of contracting Giardia infections (95% confidence interval [CI] for adjusted odds ratio [AOR] 4.53; 2.24–9.35). This finding is consistent with multiple earlier studies [48,49,51] that have emphasized the role of environmental contamination and transmission of giardiasis through the fecal-oral route. To combat this issue, interventions to improve sanitation and hygiene are essential. Previous studies have shown that latrine access, latrine use, and treating household drinking water can help reduce the odds of *G. lamblia* infection [52]. Furthermore, our research found that defecating in open areas like rivers and bushes was common in communities with inadequate toilet facilities [31]. Additionally, our study identified living in Sub-mountain valleys (95% CI for AOR 7.23; 3.51–13.74), residing near rubbish heaps (95% CI for AOR 5.52; 2.61–10.43), or living close to unpaved streets or pathways (95% CI for AOR 2.19; 1.04–4.41) as risk factors for giardiasis.

Notably, a study conducted in India reported the presence of Giardia cysts in 5% of vegetable samples [53]. Similarly, in Ethiopia, the prevalence of Giardia contamination was found in 6.9% of vegetable samples collected from local markets [54]. In Egypt, Giardia was found in 8.8% of vegetable samples [55]. Moreover, studies have demonstrated the viability of Giardia on lettuce leaves, thereby accentuating the augmented risk of contamination for consumers. Extensive research has focused on the transmission of Giardia through fecal-contaminated drinking water [56,57,58,59,60,61,62,63]. These findings underscore the heightened risk of Giardia transmission through practices such as irrigating vegetables with wastewater and fertilizing vegetable farms with manure [64,65,66]. These studies emphasize the potential for the transmission of Giardia to occur through these pathways, further highlighting the importance of addressing contamination risks in agricultural settings.

## 5. Conclusions

In conclusion, the prevalence of intestinal parasite infections in school-aged children due to their lower socioeconomic status is a significant public health concern, and our study conducted in Khyber Pakhtunkhwa, Pakistan, confirms this trend. Our findings demonstrate that various risk factors, such as cramped living conditions, consumption of tap water, unimproved latrine facilities, and low parental education and status, contribute significantly to the frequency of giardiasis. To combat this issue, integrated intervention strategies must target giardiasis prevention at both individual and community levels in Khyber Pakhtunkhwa, Pakistan. Therefore, it is essential to combine individual-level treatments, such as promoting hand washing and raising awareness about giardiasis prevention techniques, with interventions that encourage behavioral change and provide clean and appropriate water sources. Moreover, inter-sectoral cooperation between the health sector and other sectors is necessary to reduce the incidence of giardiasis in resource-poor communities effectively. Implementing these strategies will ensure a healthier future for school-aged children in Khyber Pakhtunkhwa and other regions with similar public health concerns.

## Figures and Tables

**Figure 1 children-10-01087-f001:**
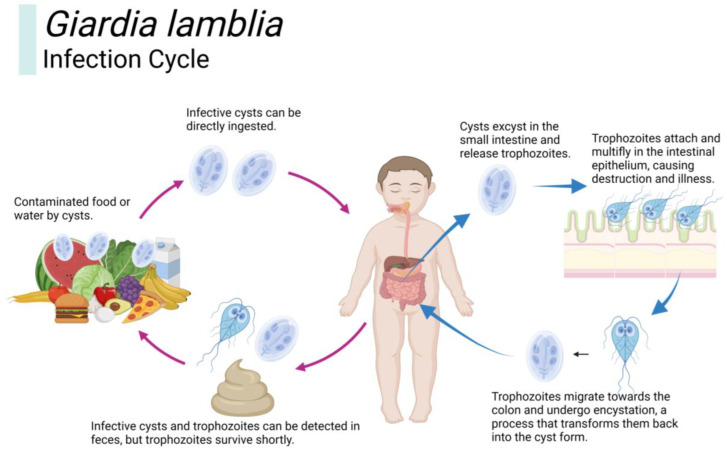
The life cycle of *Giardia lamblia.* Infective cysts are typically spread through contaminated food or water, direct ingestion, or both. Following ingestion, these cysts excyst in the human small intestine, releasing pathogenic trophozoites. The breakdown of the intestinal epithelium is induced by the trophozoites, which aid in the pathophysiology of the illness. Both cysts and trophozoites can be identified in the feces, but the discharged trophozoites have a brief existence.

**Table 1 children-10-01087-t001:** Prevalence and individual-level risk factors of giardiasis in Khyber Pakhtunkhwa, Pakistan.

Individual-Level Risk Factors	Classes	Total	Positive (% ± C.I. ^1^)
Gender	Male	428	16 (3.74 ± 1.80)
	Female	598	18 (3.01 ± 1.37)
Age group	0–6	79	1 (1.27 ± 2.47)
	7–12	86	1 (1.16 ± 2.27)
	13–24	125	10 (8.0 ± 4.76)
	25–36	288	10 (3.47 ± 2.11)
	37–48	211	7 (3.32 ± 2.42)
	49–60	237	5 (2.11 ± 1.83)
Family size	<6	293	8 (2.73 ± 1.87)
	≥6	753	26 (3.45 ± 1.30)
Mother’s educational level	Cannot read and write	457	20 (4.38 ± 1.88)
	Primary	300	9 (3.0 ± 1.93)
	Secondary	169	3 (1.78 ± 1.99)
	Higher education	100	2 (2.0 ± 2.74)
Father’s educational level	Cannot read and write	99	8 (8.08 ± 5.37)
	Primary	210	11 (5.24 ± 3.01)
	Secondary	414	10 (2.42 ± 1.48)
	Higher education	303	5 (1.65 ± 1.43)
Attend daycare	Yes	125	11 (8.8 ± 4.97)
institution/Kindergarten	No	901	23 (2.55 ± 1.03)
Handwashing behavior of	Washes at critical times	750	15 (2.0 ± 0.1)
mothers	Does not wash at critical times	276	19 (6.88 ± 2.99)
Companion animal	Yes	642	31 (4.83 ± 1.66)
	No	384	3 (0.78 ± 0.88)
Stray dogs/cats enter home	Yes	275	21 (7.64 ± 3.14)
	No	751	13 (1.73 ± 0.93)

Abbreviations: ^1^ C.I.: 95% confidence interval.

**Table 2 children-10-01087-t002:** Prevalence and community-level risk factors of giardiasis in Khyber Pakhtunkhwa, Pakistan.

Community-Level Factors	Classes	Total	Positive (% ± C.I. ^1^)
Agro-ecological zones	High dry mountains	114	5 (4.39 ± 3.76)
	Sub mountain valleys	114	7 (6.14 ± 4.41)
	Sub humid mountains	114	4 (3.51 ± 3.38)
	Wet mountains	114	4 (3.51 ± 3.38)
	Valley plains	114	5 (4.39 ± 3.76)
	Piedmont plain	114	2 (1.75 ± 2.41)
	Semi-arid piedmont	114	3 (2.63 ± 2.94)
	Western mountains	114	3 (2.63 ± 2.94)
	Desert plains	114	1 (0.88 ± 1.71)
Drinking water sources	Piped	846	21 (2.48 ± 1.05)
	Un-piped	180	13 (7.22 ± 3.78)
Drinking water boiled or filtered	Yes	75	1 (1.33 ± 2.60)
	No	951	33 (3.47 ± 1.16
Domestic water storage vessel	Adequate	767	10 (1.30 ± 1.16)
	Inadequate	259	24 (9.27 ± 3.53)
Presence of rubbish heap near the	Not present	780	12 (1.54 ± 0.86)
house	Present	246	22 (8.94 ± 3.57)
Condition of street or pathway	Paved	711	13 (1.83 ± 0.98)
	Unpaved	515	21 (4.08 ± 1.71)
Latrine facilities	Improved	800	15 (1.88 ± 0.94)
	Unimproved	226	19 (8.41 ± 3.62)

Abbreviations: ^1^ C.I.: 95% confidence interval.

**Table 3 children-10-01087-t003:** Identified individual level risk factors of giardiasis in Khyber Pakhtunkhwa, Pakistan.

Empty Model Factors	Classes	Models		
		Model I	Model II	Full Model
		AOR (95 ± C.I.)	AOR (95 ± C.I.)	AOR (95 ± C.I.)
Age group	0–6	0.60 (0.29–1.4)	-	0.8 (0.29–1.4) *
(months)	7–12	0.55 (0.26–1.3)	-	0.61 (0.26–1.3)
	13–24	3.81 (1.82–7.71)	-	3.92 (1.82–7.71)
	25–36	1.65 (0.81–3.41)	-	1.81 (0.81–3.41)
	37–48	1.58 (0.81–3.22)	-	1.62 (0.81–3.22)
	49–60	1.00	-	1.00
Family size	<6	1.00	-	1.00
	≥6	1.19 (0.584–2.41)	-	1.0 (0.51–1.91)
Mother’s	Cannot read and write	2.19 (0.98–4.41)	-	2.22 (1.11–4.52) *
educational	Primary	1.50 (0.81–3.12)	-	1.38 (0.66–3.18)
level	Secondary	0.89 (0.56–1.82)	-	0.89 (0.56–1.82)
	Higher education	1.00	-	1.00
Father’s	Cannot read and write	4.75 (2.21–9.58)	-	4.82 (2.33–9.61) *
educational	Primary	3.08 (1.67–6.07)	-	3.23 (1.69–6.14)
level	Secondary	1.42 (0.65–2.92)	-	1.38 (0.59–3.1)
	Higher education	1.00	-	1.00
Attend daycare	Yes	3.38 (1.72–6.65)	-	3.12 (1.61–6.42) *
institution/Kindergarten	No	1.00	-	1.00
Handwashing behavior	Washes at critical times	1.00	-	1.00
	Does not wash at critical times	3.44 (1.72–6.82)	-	3.61 (1.91–6.94) *
Companion animal	Yes	6.04 (2.96–12.12)	-	5.72 (2.81–11.78) *
	No	1.00	-	1.00
Stray dogs/cats	Yes	4.49 (2.10–8.84)	-	4.62 (2.23–8.92) *
enter home	No	1.00	-	1.00

* Significant at *p* value < 0.05, C.I.; Confidence interval, AOR; Adjusted odds ratio.

**Table 4 children-10-01087-t004:** Identified community-level risk factors of giardiasis in Khyber Pakhtunkhwa, Pakistan.

Empty Model Factors	Classes	Models		
		Model I	Model II	Full Model
		AOR (95 ± C.I.)	AOR (95 ± C.I.)	AOR (95 ± C.I.)
Agro-ecological zones	High dry mountains	-	4.90 (2.43–9.81)	5.11 (2.46–9.92) *
	Sub mountain valleys	-	6.97 (3.47–13.63)	7.23 (3.51–13.74)
	Sub humid mountains	-	4.12 (2.12–8.57)	3.91 (2.03–8.41)
	Wet mountains	-	4.12 (2.12–8.57)	3.91 (2.03–8.41)
	Valley plains	-	4.90 (2.43–9.81)	5.11 (2.46–9.92)
	Piedmont plain	-	2.13 (1.03–4.28)	1.92 (0.98–4.02)
	Semi-arid piedmont	-	2.94 (1.31–5.87)	3.14 (1.37–5.99)
	Western mountains	-	2.94 (1.31–5.87)	3.14 (1.37–5.99)
	Desert plains	-	1.00	1.00
Drinking water sources	Piped	-	1.00	1.00
	Un-piped	-	2.91 (1.42–5.98)	3.30 (1.5–6.8) *
Domestic water	Adequate	-	1.00	1.00
storage vessel	Inadequate	-	7.13 (3.12–13.15)	6.5 (3.06–12.88) *
Drinking water boiled	Yes	-	1.00	1.00
or filtered	No	-	2.61 (1.21–5.41)	2.88 (1.46–6.12) *
Presence of rubbish	Not present	-	1.00	1.00
heap near the house	Present	-	5.81 (2.78–10.94)	5.52 (2.61–10.43) *
Condition of street	Paved	-	1.00	1.00
or pathway	Unpaved	-	2.23 (1.09–4.51)	2.19 (1.04–4.41) *
Latrine facilities	Improved	-	1.00	1.00
	Unimproved	-	4.47 (2.18–9.21)	4.53 (2.24–9.35) *

* Significant at *p* value < 0.05, C.I.; Confidence interval, AOR; Adjusted odds ratio.

## Data Availability

The datasets generated and/or analyzed during the current study are not publicly available due to data transcripts, including personal participant information, not suitable for sharing but could be available from the corresponding authors at reasonable request.

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
