# Peer review of "Individual and Community-Level Risk Factors for Giardiasis in Children under Five Years of Age in Pakistan: A Prospective Multi-Regional Study"

_children, 2023, doi:10.3390/children10061087_

Round 1
Reviewer 1 Report
The aims of this work is to estimate the prevalence of Giardia lamblia
infection and identify the associated risk factors at both individual and community levels in a pediatric population in different agroecological zones of Khyber Pakhtunkhwa, Pakistan. By using stratified sampling, 1026 households were recruited from nine agroecological zones. Stool samples were collected from 1026 children up to the age of five years and processed for detection of Giardia.
Data on potential risk factors were collected using a pre-structured questionnaire. A multivariable logistic regression model was used to identify risk factors associated with giardiasis. The authors found that the prevalence of giardiasis in the study area was 3.31%. Children aged 13-24 months were
found to be at higher risk for giardiasis.
A lifestyle whitout taking care of the house cleanliness or the environment
is the main factor of Giardia lamblia infection . The unfiltered and dirty
water were identified as a risk factor that infect children with Giardia lamblia
infection.
I found this work significant and meaningful. But I have two questions
for the authors.
1. The Authors are invited to explain their choice of stratified sampling.
Why not, for example, simple random sampling, multi-stage sampling,
cluster sampling, etc.?
2. The authors are also invited to provide further details on the questionnaire
used.
No comment
Author Response
Response letter to reviewer 1
Reviewer 1:
- The Authors are invited to explain their choice of stratified sampling. Why not, for example, simple random sampling, multi-stage sampling, cluster sampling, etc.?
Reply:
Thank you for your valuable comments on our manuscript. We appreciate your feedback and would like to address your query regarding the choice of stratified sampling.
Our study aimed to determine the prevalence of G. lamblia infection in an apparently healthy pediatric population across several agro-ecological zones (AEZs) of Khyber Pakhtunkhwa, Pakistan, and identify any potential individual and community-level risk factors. To achieve this, we aimed to collect an equal number of samples from each AEZ, ensuring that the samples are representative.
Considering the objective mentioned above, we chose to employ stratified sampling by treating each AEZ as a stratum. This approach allowed us to randomly select samples within each stratum, ensuring that we have an equal representation from each AEZ.
If we had used simple random sampling, there would have been a possibility of under or over-representation of any of the nine AEZs, which could have biased our results. Similarly, multi-stage sampling would have only covered a subset of the groups or clusters, leaving some AEZs underrepresented.
To clarify our choice of stratified sampling, we have explained it in the "Materials and Methods" section of the revised manuscript. We stated, "Each zone was treated as a stratum, and the stratified sampling technique was employed to select the study participants so as to collect an equal number of samples from each AEZ and ensure the samples are representative of their respective AEZs.".
- The authors are also invited to provide further details on the questionnaire used.
Reply:
Thank you for your comments. The questionnaire form has been added to the supplementary file. Moreover, further details on the questionnaire used have been added in the section “Questionnaire and observation checklist for assessing giardiasis risk factors” as follows: “In this study, we employed a comprehensive questionnaire and observation checklist to assess the risk factors associated with giardiasis at both the individual and community levels. The observation checklist was designed to gather data on various environmental factors, including domestic water storage vessels, the presence of rubbish heaps near homes, the condition of streets or pathways, latrine facilities, and the presence of companion animals. Simultaneously, the questionnaire focused on evaluating individual-level risk factors. It encompassed a range of factors, such as child gender, age, family size, mother's and father's educational level, attendance at daycare or kindergarten, handwashing behavior, and the potential entry of stray dogs and cats into the home. To ensure accuracy and reliability, face-to-face interviews were conducted with guardians or primary caregivers to gather data on these specific risk factors. Furthermore, community-level risk factors were also assessed as part of our investigation. These included the agro-ecological zones (AEZs) in the study area, drinking water sources, the usage of boiled or filtered drinking water, domestic water storage vessels, the presence of rubbish heaps near homes, the condition of streets or pathways, and the availability and quality of latrine facilities. By using this comprehensive questionnaire and observation checklist, we aimed to gather a detailed understanding of the risk factors associated with giardiasis at both the individual and community levels. This approach allowed us to capture a broad range of variables and factors that could potentially contribute to the prevalence of giardiasis in the studied population.”.

Reviewer 2 Report
Undoubtedly, water and personal hygiene (hand washing) are crucial to the spread of Giardiasis. But the other way of infection is the consumption of vegetables (not washed well or washed with water contaminated with human feces). Such data would explain why small children are also infected. Both contaminated water and poor hygiene are responsible for the high prevalence of this parasitosis. But the main reason for the spread of Giardiasis is defecation in the environment, the use of feces for fertilization and the lack of sanitation. These data can be supplemented in the text. Preventive examinations of people and radical cure of the infected is a major factor in reducing the number of infected.
I have a question: Are the people ЕLISA positive for Giardiasis examined by the microscopic method of fecal sample for the presence of cysts and are they treated? Confirmation with the microscopic method is of great importance for diagnosis.
Author Response
Response letter to Reviewer 2
Reviewer 2:
1. Undoubtedly, water and personal hygiene (hand washing) are crucial to the spread of Giardiasis. But the other way of infection is the consumption of vegetables (not washed well or washed with water contaminated with human feces). Such data would explain why small children are also infected. Both contaminated water and poor hygiene are responsible for the high prevalence of this parasitosis. But the main reason for the spread of Giardiasis is defecation in the environment, the use of feces for fertilization and the lack of sanitation. These data can be supplemented in the text. Preventive examinations of people and radical cure of the infected is a major factor in reducing the number of infected.
Reply:
Thank you for your professional comments. We totally agree with you. Regarding to the consumption of vegetables as an important risk factor for Giardiasis is well discussed in the discussion section as follows:
“Notably, a study conducted in India reported the presence of Giardia cysts in 5% of vegetable samples [53]. Similarly, in Ethiopia, the prevalence of Giardia contamination was found in 6.9% of vegetable samples collected from local markets [54]. In Egypt, Giardia was found in 8.8% of vegetable samples [55]. Moreover, studies have demonstrated the viability of Giardia on lettuce leaves, thereby accentuating the augmented risk of contamination for consumers. Extensive research has focused on the transmission of Giardia through fecal-contaminated drinking water [56-63]. These findings underscore the heightened risk of Giardia transmission through practices such as irrigating vegetables with wastewater and fertilizing vegetable farms with manure [64-66]. These studies emphasize the potential for the transmission of Giardia to occur through these pathways, further highlighting the importance of addressing contamination risks in agricultural settings.”.
2. I have a question: Are the people ЕLISA positive for Giardiasis examined by the microscopic method of fecal sample for the presence of cysts and are they treated? Confirmation with the microscopic method is of great importance for diagnosis.
Reply:
Thank you for your question and valuable input. We appreciate your interest in our study. To address your query, individuals who tested positive for Giardiasis through the ELISA method were not examined using the microscopic method for the presence of cysts. As the study was purely observational in nature, we did not provide treatment to the participants. However, it is important to note that they were advised to consult a healthcare professional for proper diagnosis and appropriate treatment. We acknowledge that confirmation with the microscopic method is indeed crucial for accurate diagnosis. Given the scope and design of our study, we focused on assessing the prevalence of Giardiasis and identifying potential risk factors among the study population. The objective was to generate epidemiological insights rather than providing individual-level diagnosis or treatment.
